# High Consumption of Sugar-Sweetened Beverages Is Associated with Low Bone Mineral Density in Young People: The Brazilian Birth Cohort Consortium

**DOI:** 10.3390/nu15020324

**Published:** 2023-01-09

**Authors:** Maylla Luanna Barbosa Martins Bragança, Eduarda Gomes Bogea, Poliana Cristina de Almeida Fonseca Viola, Juliana dos Santos Vaz, Susana Cararo Confortin, Ana Maria Baptista Menezes, Helen Gonçalves, Heloisa Bettiol, Marco Antonio Barbieri, Viviane Cunha Cardoso, Antônio Augusto Moura da Silva

**Affiliations:** 1Postgraduate Programme in Collective Health, Federal University of Maranhão, São Luís 65020-070, Maranhão, Brazil; 2Department of Nutrition, Federal University of Piauí, Teresina 64049-550, Piauí, Brazil; 3Postgraduate Programme in Nutrition and Food, Faculty of Nutrition, Federal University of Pelotas, Pelotas 96010-610, Rio Grande do Sul, Brazil; 4Postgraduate Programme in Epidemiology, Department of Social Medicine, Faculty of Medicine, Federal University of Pelotas, Pelotas 96020-220, Rio Grande do Sul, Brazil; 5Postgraduate Programme in Child and Adolescent Health, University of São Paulo, Ribeirão Preto 14048-900, São Paulo, Brazil

**Keywords:** beverages, sugar, bone mineral density, young adult

## Abstract

Sugar-sweetened beverages (SSB) consumption may be associated with a reduction in bone mineral density (BMD). The aim of this study was to evaluate the association between the consumption of SSB and BMD among young people. We performed a cross-sectional study that evaluated 6620 young people (18–23 years of age) from three Brazilian birth cohorts (Ribeirão Preto, Pelotas, and São Luís). We analyzed the daily frequency and the amount and energy contribution of the SSB, which were obtained through a food frequency questionnaire. Total body and lumbar spine BMD (g/cm^2^), measured by dual-energy X-ray absorptiometry, were the outcomes. Unadjusted linear regression models, adjusted for sex, socioeconomic class, physical activity, smoking, alcohol consumption, and body mass index were used. The highest tertile of SSB consumption frequency (2.1–16.7 times/day) was associated with a lower lumbar spine BMD (β = −0.009; 95% CI: −0.017; −0.001; standardized β = −0.03). This association persisted after adjustment for confounders (β = −0.008; 95% CI: −0.016; −0.001; standardized β = −0.03). No association was observed between SSB consumption frequency and total body BMD or between the amount and energy contribution of SSB and total body or lumbar spine BMD. A high frequency of SSB consumption was associated with a low lumbar spine BMD.

## 1. Introduction

Bone mineral density (BMD) is defined as the concentration of bone tissue in a given volume of bone and has been widely studied in older adults because of the high prevalence of osteopenia, osteoporosis and fractures resulting from the loss of bone minerals. However, its study in young people is important, since the BMD of adults and older adults depends on the bone mass acquired up to 30 years of age [1]. It is therefore important to know the factors that interfere with BMD in young people in order to reduce problems resulting from bone demineralization at older ages [1].

Intrinsic factors such as genetic and hormonal factors, sex, and ethnicity are responsible for about 80% of bone mass acquisition. In addition, extrinsic factors such as mechanical, physical activity, and nutritional factors interfere with BMD [2,3]. With respect to the nutritional factors, a high intake of milk, milk products, calcium, magnesium, phosphorus, and vitamin D is related to a great bone increment, while a high intake of foods containing caffein, phosphates, and sodium is associated with a low BMD [4]. Sugar-sweetened beverages (SSB) consumption has also been associated with a reduction in BMD [5]. This food group includes any beverages that contain added sucrose or high-fructose corn syrup [6].

Studies in children, adolescents, or young adults evaluated the associations between SSB and BMD [5,7,8]. Those studies investigated the influence of the consumption of different SSB on BMD, some beverages including fruit and vegetable juice or milk that contain nutrients and may have a beneficial effect on bone mineralization [5,7,8]. However, there are no studies that evaluated the exclusive consumption of industrially sweetened beverages and their association with BMD.

Industrially sweetened beverages such as soft drinks, juices, and chocolate drinks can have a greater impact on bone demineralization than other beverages sweetened by individuals, such as milk and fruit juices, which also contain vitamins and minerals. These industrially sweetened beverages contain large amounts of added sugars as the main sources of dietary sugars and generally contain corn syrups [9,10]. These syrups are characterized by a high concentration of fructose, whose metabolic and physiological mechanisms further contribute to bone demineralization [9,10].

The consumption of these beverages is frequent among young Brazilians, with the prevalence of SBB consumption ranging from 30% to 37%, according to data from national surveys. Therefore, considering the increase in the consumption of SSB among young people [11,12,13] and the possible deleterious effect of the consumption of these beverages on BMD [9,10], the objective of this study was to evaluate the association between SSB consumption and BMD in young people aged 18 to 23 years of age from three cohorts of Brazilian birth. The hypothesis of this study was that SSB consumption is associated with a reduction in total body and lumbar spine BMD.

## 2. Methods

### 2.1. Study Design and Sample

This was a cross-sectional study using data from three Brazilian birth cohorts from the cities of Ribeirão Preto (São Paulo), Pelotas (Rio Grande do Sul), and São Luís (Maranhão), obtained from the RPS Brazilian Birth Cohorts Consortium (Ribeirão Preto, Pelotas and São Luís) [14]. The three cohorts included liveborn infants from hospital deliveries of mothers resident in the urban area of these cities. The cohorts were started in 1993 in Pelotas, in 1994 in Ribeirão Preto, and in 1997/1998 in São Luís. This study used the follow-up data obtained in Pelotas in 2015/2016 considering participants from 21 to 23 years of age, in Ribeirão Preto in 2016/2017, with participants of 22 and 23 years of age, and in São Luís in 2016, with participants of 18 and 19 years of age. More details of these cohorts are described in other studies [14,15,16].

The São Luís cohort evaluated 2542 children at birth. Of these, 654 subjects participated in the new data collection at 18 and 19 years of age. In view of the difficulty in locating the participants of the original birth cohort, in order to increase the power of the sample and to prevent future losses, the sample size was increased by including 1861 subjects born in São Luís in 1997 who did not participate in the original birth cohort, thus totaling 2515 adolescents. Of these, 122 subjects with missing data of BMD or SSB consumption were excluded, resulting in 2393 young people from São Luís. The Ribeirão Preto cohort evaluated 2839 children at birth and 622 subjects of 22 and 23 years of age. To increase the sample size of the study, 419 subjects born in Ribeirão Preto in 1994 who were not selected at first to participate in the birth cohort were included, totaling 1041 young adults. One hundred and twenty-six subjects with missing data of BMD or SSB consumption were excluded, resulting in a sample of 915 young people from Ribeirão Preto. The Pelotas cohort evaluated 5249 subjects at birth and 3617 subjects of 22 and 23 years of age. Of these, 305 subjects with missing data of BMD or SSB consumption were excluded, resulting in 3312 subjects evaluated in Pelotas. Therefore, the final sample of this study consisted of 6620 young people from the three birth cohorts of the participating cities.

The data were collected by trained professionals. Sociodemographic characteristics, life habits, and food intake data were obtained with standardized questionnaires. The Research Electronic Data Capture (Redcap^®^) software was used for online data capture and management [17].

### 2.2. Food and Sugar-Sweetened Beverage Intake

The SSB were considered the exposure variables. Non-diet soft drinks, processed juices, and chocolate drinks were evaluated. All of them are industrially sweetened beverages that contain high amounts of added sugars [9]. Milk, coffee, and natural juices were not included because these beverages contain minerals, vitamins, and caffeine that can affect bone density. Three indicators of SSB consumption were studied: daily frequency, amount consumed in milliliters, and the contribution of energy of SSB in relation to total calories. These three indicators were categorized into tertiles.

The intake of food and SSB was evaluated using a food frequency questionnaire (FFQ) that measures consumption over the last 12 months. The FFQ developed by Schneider et al. [18] and adapted and validated by Vaz et al. [19] was used in Pelotas. The same FFQ was applied in Ribeirão Preto and São Luís; however, the questionnaire was adapted to regional dietary habits. Further details on FFQ validation, collection and analysis have been described elsewhere [20,21].

The frequency of SSB consumption and food items from FFQ was obtained using the following response options: never or <1 time/month; 1 to 3 times/month; 1 time/week; 2 to 4 times/week; 5 to 6 times/week; 1 time/day; 2 to 4 times/day; ≥5 times/day [22]. The frequency of daily SSB consumption was first estimated by the conversion of the frequency reported for each item to the annual consumption in order to capture less frequent intakes. The frequency of <1 time/month was classified as absent consumption, that of 1 to 3 times/month was transformed to 12.0 times/year, that of 1 time/week was transformed to 52.0 times/years, that of 2 to 4 times/week was transformed to 104.0 times/year, that of 5 to 6 times/week was transformed to 260.0 times/year, that of 1 time/day was transformed to 365.25 times/year, that of 2 to 4 times/day was transformed to 730.5 times/year, and that of ≥5 times/day was transformed to 1826.25 times/year. Then, the annual frequency was converted to daily frequency by dividing it by 365.25.

The amounts of SSB and the food items in the FFQ consumed were obtained based on the size of the portion. For this purpose, at the time of application of the FFQ, photographs with average portion sizes of each food were made available for viewing on a computer to minimize memory bias and improve the quality of the information. We then recorded whether the participant consumed the amount of the portions viewed (corresponding to the average portion) or a higher (1.5 times the average portion) or lower amount (0.5 times the average portion) [18]. The amount in milliliters of the average portion of the foods was obtained using the Brazilian reference for the Evaluation of Household Measures of Food Consumption [23]. A 240 mL glass of a non-diet soft drink, juice box, or powder and chocolate drink was considered an average portion. To calculate the daily SSB consumption in milliliters, the daily consumption frequency was multiplied by the value in milliliters of the average portion (240 mL), the largest portion recorded (1.5 times the average portion), or the smallest portion recorded (0.5 times the average portion).

For the calculation of the energy contribution of SSB, the daily dietary intake in grams or milliliters was first converted to the amount of macronutrients according to the Brazilian Food Composition Table (TACO), the Table of Nutritional Composition of Food Products Consumed in Brazil, the USDA National Nutrient Database for Standard Reference, or the information on food labels [24,25,26]. The energy intake from each food was estimated by multiplying the carbohydrate and protein values by 4 kcal and the lipid values by 9 kcal. The daily energy intake from each food was then obtained after summing the calories from the macronutrients. The total daily energy intake was calculated by summing the calories consumed from all food items of the FFQ. The energy contribution of SSB was then determined in relation to total calories.

### 2.3. Bone Mineral Density (BMD)

The BMD of the participants was measured by dual-energy X-ray absorptiometry (DXA) using a Lunar Prodigy enCORE-based densitometer (GE Healthcare^®^, Chicago, IL, USA). The DXA device was calibrated daily. For the examination, the participants were barefoot and used light, tight-fitting clothing without earrings, rings, dentures, and other types of metal materials. The measurements were performed at two sites, total body and lumbar spine, according to the Brazilian Society of Clinical Densitometry. Femoral measurements were not used since this bone continues to grow until 20 years of age [2]. Bone mineral density was determined as the ratio between bone mineral content (g) and bone area (cm^2^). The two measurements in g/cm^2^ were considered the outcome variables.

### 2.4. Theoretical Model

The theoretical model including the exposure, outcome, and confounding variables is represented by a directed acyclic graph (DAG) designed with the DAGitty 3.2 program [27], in which the causal relationships are given by unidirectional arrows to show the direction of causality (Figure 1). Each variable in the DAG is represented by a square with a different meaning: “►” indicates the exposure, “I” the outcome, and all other variables are confounders.

After the development of the DAG, a minimum set of adjustment variables established based on the backdoor path criterion was selected to control for confounding factors and was used to adjust the data in order to avoid unnecessary adjustments, spurious associations, and estimation errors [28]. The backdoor path criterion considers the need to adjust for variables that are common causes of both the exposure and the outcome and variables that precede the confounders. Adjustments for mediators (which would block causal flow and suppress the effect of exposure on outcome), colliders (which would cause bias since colliders block the flow of a spurious association between two variables), and descendants of colliders are not suggested [29,30].

After the analysis of the theoretical model through the application of the backdoor path criterion, the minimum set of adjustment variables suggested to control for confounders were sex, socioeconomic class, alcohol consumption, smoking, and physical activity. Age, skin color, height, lean mass, fat mass, stress, and the intake of milk, protein, sugar, caffein, vitamin V12, vitamin C, vitamin D, calcium, and sodium were included in the DAG as confounders [4,25] but were not considered in the analysis since they were not identified in the minimum set of adjustment variables.

### 2.5. Covariates

The following adjustment variables were considered: sex (female; male); socioeconomic classification according to the criteria of the Brazilian Association of Research Companies (ABEP in the Portuguese acronym) categorized into classes A, B, C, and D/E (where class A includes the richest and most educated, and class D/E comprises the poorest and least educated). In 2016, the average monthly household income of each socioeconomic class was A = USD 5989.00, B = USD 2021.00, C = USD 1241.00, and D/E = USD 220.00 [31]; we also evaluated the current alcohol consumption (no or yes, as the habit of consuming alcoholic beverages at least once a month); current smoking (no or yes, as the habit of cigarette smoking at least once a week); physical activity level (insufficiently active: less than 150 min of physical activity per week; physically active: 150 min or more of physical activity per week). In São Luís, physical activity was evaluated using the Self-Administered Physical Activity Checklist (SAPAC) [32]. In Ribeirão Preto and Pelotas, the International Physical Activity Questionnaire (IPAQ) was used [33].

The body mass index (BMI) was assessed as a mediator, in case adjusting for this variable would make the association between SSB and BMD disappear [34]. This variable was evaluated as a mediator since a high SSB consumption can increase the BMI which, in turn, is related to a high BMD [35]. The BMI was calculated as weight in kg divided by height in meters squared. The height (cm) of the adolescents was measured with a stadiometer (Altura Exata^®^, Minas Gerais, Brazil), and the body weight (kg) with a Filizola^®^ scale.

### 2.6. Statistical Analysis

The data were analyzed using the STATA^®^ 14.0 software (StataCorp, College Station, TX, USA). For descriptive analysis, absolute and relative frequencies are reported for categorical variables, and mean and standard deviation for continuous variables.

The associations between SSB consumption and BMD were analyzed using β coefficients and 95% confidence intervals in unadjusted linear regression and adjusted for the minimum set of variables identified by the theoretical model and for BMI. A level of significance of 0.05 was adopted. The standardized β coefficient was used to assess the effect size of the associations studied in order to understand their practical and clinical significance. This coefficient indicates the standard deviation change in the outcome variable caused by the change of one standard deviation in the exposure variable. A coefficient of about 0.10 indicated a small effect size [36]. The effects of the interactions between city of residence and SSB and between sex and SSB on total body and lumbar spine BMD were tested. These interactions were nonsignificant and were therefore included in the same model (city and sex). The level of significance for the interaction analysis was set at 0.10.

### 2.7. Ethical Criteria

The studies were approved by the Ethics Committees of the respective institutions: in São Luís, by the Ethics Committee of the University Hospital of the Federal University of Maranhão (Approval number 1.302.489); in Ribeirão Preto, by the Ethics Committee of the University Hospital of the Ribeirão Preto Medical School, University of São Paulo (Approval number 2.998.903), and in Pelotas, by the Ethics Committee of the Medical School, Federal University of Pelotas (Approval number 1.250.366). All participants in the cohorts signed a free informed consent form.

## 3. Results

The sample predominantly consisted of young females (53.8%) belonging to socioeconomic class C (41.3%), who were non-smokers (89.1%), consumed alcoholic beverages (61.3%), and practiced physical activity (63.4%) (Table 1).

The mean total body and lumbar spine BMD was 1.2 g/cm^2^. The mean frequency of SSB consumption was 1.6 times/day, with a daily consumption of 281.5 mL, which provided 5.9% of total dietary calories. The participants in the highest tertile of SSB consumption consumed these beverages on average 3.4 times/day, with a daily consumption of 629.0 mL, which provided 11.4% of the total dietary calories (Table 2).

An association was found between the highest tertile of daily SSB consumption frequency (2.1 to 16.7 times/day) and lumbar spine BMD (β = −0.009; 95% CI: −0.017; −0.001). The effect size of this association was small (standardized β = −0.03). This association persisted after adjustment for confounding variables (β = −0.012; 95% CI: −0.021; −0.004; standardized β = −0.042) and for BMI (β = −0.008; 95% CI: −0.016; −0.001; standardized β = −0.03). There was no association between the highest tertile of daily SSB consumption frequency and total body BMD (β = −0.003; 95% CI: −0.009; 0.004; standardized β = −0.12) (Table 3).

No associations were observed between the highest daily amount of SSB consumption (275.6 to 3862.5 mL/day) and total body (β = 0.008; 95% CI: 0.002; 0.015; standardized β = 0.025) or lumbar spine BMD (β = 0.001; 95% CI: −0.07; 0.009; standardized β = 0.005) (Table 4 and Table 5), nor between the highest energy contribution of SSB (6.9% to 25.7% of daily calories) and total body (β = 0.004; 95% CI: −0.002; 0.011; standardized β = 0.02) or lumbar spine BMD (β = −0.001; 95% CI: −0.009; 0.007; standardized β = −0.003) (Table 5).

## 4. Discussion

The main result of this study involving a sample of young Brazilians is the association between a higher frequency of daily SSB consumption and lower lumbar spine BMD. This association was independent of sex, socioeconomic class, alcohol consumption, smoking, physical activity level, and BMI.

We found no associations between the frequency of daily SSB consumption and the total body BMD or between the amount consumed and the energy contribution of SSB and the total body or lumbar spine BMD. However, despite the small magnitude of the effect, a higher frequency of SSB consumption was associated with a lower lumbar spine BMD. A change of one standard deviation in the SSB consumption frequency reduced the lumbar spine BMD by 0.03 standard deviations. The high consumption of SSB by these adolescents probably did not result in a marked bone demineralization because this is a young population in which bone mineralization is still expected to occur [37]. This fact may also explain the lack of associations between the frequency of daily SSB consumption and the total body BMD and between the amount consumed and energy contribution of SSB and the total body and lumbar spine BMD. Furthermore, these results are important because the observation of an association only with lumbar spine BMD may indicate the onset of bone demineralization that could spread throughout the body if excessive consumption of SSB continues in the sample studied over the years.

Other studies have reported similar results with respect to the association between SSB and BMD. In a sample of German children and adolescents aged 6 to 18 years, Libuda, et al. [5] showed that the consumption of SSB such as soft drinks, iced tea, fruit juice, fruit-flavored drinks, and sports and energy drinks was associated with a reduction of 0.009 mg/mm^2^ in BMD. Whiting et al. [7] found that the consumption of soft drinks and fruit juices by 112 Canadian adolescents was associated with a lower bone mass, with a reduction of up to 0.256 g among those consuming higher amounts of SSB. Studying 750 adult Puerto Ricans (47–79 years) living in Massachusetts, Akinlawon et al. [38] observed that the consumption of SSB including soft drinks, fruit drinks and nectar, apple juice, 100% juice blends, sweetened tea, energy drinks and coffee beverages was associated with a reduction of 0.001 g/cm^2^ in lower femoral neck BMD. However, the authors found no association for other bone sites such as the trochanter, total hip, or lumbar spine.

Different results including indirect associations between SSB and BMD have been reported in other studies. Bukhar et al. [39] studied 267 female university students in Saudi Arabia aged 19 to 24 years and found an association between obesity and high BMD, with obesity being associated with a greater intake of coffee, tea, soft drinks, milk, and juices in a separate analysis. Using data from NANHES II that included subjects aged 20 to 35 years, Braun [8] observed that a high consumption of SSB reduced milk intake, which was associated with a lower BMD. The association between SSB and BMD is believed to be due not only to lower milk intake or to the impact of obesity on BMD as reported in other studies, but also to the presence of added sugars in these beverages, which can trigger physiological and metabolic mechanisms that result in bone demineralization [8,39].

According to some studies, SSB consumption can cause hyperinsulinemia which, in turn, produces hypercalciuria by inhibiting the renal tubular reabsorption of calcium, reducing calcium availability for bone mineralization [10,40,41]. Another possible mechanism would be the inhibition of osteoblast proliferation when hyperglycemia, caused by the consumption of SSB, compromises the response of osteoblasts to insulin-like growth factor I (IGF-I), reducing the proliferation of these cells and impairing osteogenesis [42].

It is noteworthy that the frequency of daily SSB consumption is a greater risk factor for a lower BMD than the SBB amount consumed or energy contribution, as the frequency of daily consumption has another physiological effect that can increase bone demineralization. Thus, the more times a day SSB are consumed, the longer the duration of hyperglycemia [43], since added sugars exert an effect on postprandial hyperglycemia that can trigger a proinflammatory cascade and may persist for up to 16 h after the hyperglycemic peak. These effects of hyperglycemia can further contribute to hypercalciuria, reducing the activity of osteoblasts and consequently bone synthesis [37,38].

In the present study, we decided not to include the intake of milk, coffee, and natural juices because, although they may contain added sugars, some components of these beverages such as minerals, vitamins, or caffeine can increase BMD, which would be an effect contrary to the objective of this study. Only the associations between industrially sweetened beverages and BMD were studied in the young adults evaluated here. This decision was based on the type and quantity of sugars used in industrially sweetened beverages and their great negative health impact [9]. Sucrose and corn syrup, which contains a high concentration of fructose, are the sugars most widely used by the industry to sweeten SSB because of their low cost. Excessive fructose intake can trigger another mechanism that can contribute to a low BMD, in which the catabolism of fructose in the liver results in acidosis through the formation of uric acid, and calcium ions from the bones are thus diverted to the blood to restore body homeostasis [44,45].

The strengths of this study include the use of DXA as an accurate method to estimate BMD, DMB measurements at two bone sites (total body and lumbar spine), and three indicators of SSB consumption (daily frequency, amount, and energy contribution) for a more detailed analysis of the associations between SSB and BMD. In addition, the study adjusted a minimum set of confounders in order to avoid bias, unnecessary adjustments, spurious associations, and estimation errors in the association between SSB consumption and BMD.

As a limitation, although this study comprised data from three birth cohorts, only cross-sectional data were used in the analysis, a fact that impeded the determination of causality. Despite this limitation, the questionnaire used to assess SSB consumption referred to the last year, representing the habitual intake. It should be noted that FFQ may overestimate food intake. However, the instrument is based on the principle that the estimated habitual dietary intake is a more important exposure factor than the focal intake and is therefore preferable in studies evaluating the intensity of an exposure [46]. Another limitation is the geographical difference of the study populations. São Luis is a metropolitan city in the cost in the northeast of Brazil and has a tropical climate, Ribeirao Preto is in the southwest, with an average temperature of 23 °C throughout the year, and Pelotas is in the extreme south of the country, with a winter season and temperatures ranging from −5 to 15 °C. Taking into account that, the city of residence was considered in adjustments to control for geographical differences. Another limitation refers to the fact that the family history of bone health, such as the presence of osteoporosis, was not considered.

## 5. Conclusions

A high SSB consumption was associated with a lower lumbar spine BMD but not with total body BMD. Despite the small magnitude of this reduction, this result is a matter of concern because of the young age of the population studied, since the negative effect of SSB consumption on BMD was observed during a phase of life when bone mass formation is still occurring. Thus, associations of greater magnitude between SSB and BMD may occur at older ages if this population continues to consume SSB in an excessive and persistent manner.

## Figures and Tables

**Figure 1 nutrients-15-00324-f001:**
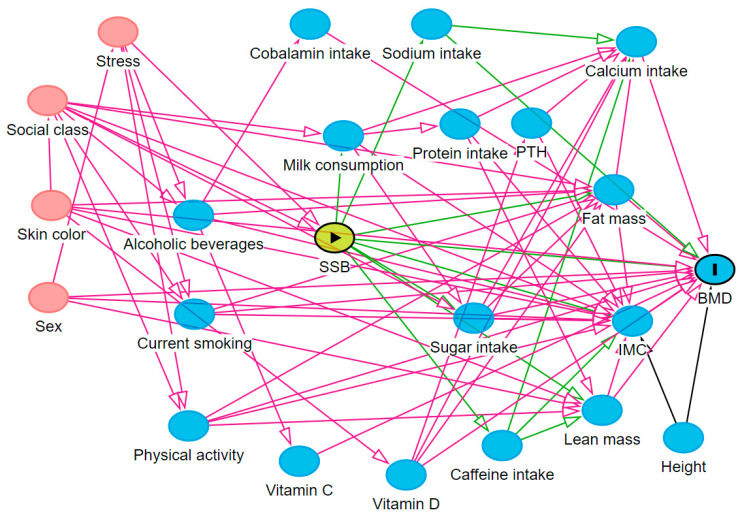
Directed acyclic graph of the association between the consumption of sugar-sweetened beverages (exposure) and bone mineral density (outcome).

**Table 1 nutrients-15-00324-t001:** Sociodemographic characteristics and life habits of young people of the RPS Brazilian Birth Cohorts Consortium (Ribeirão Preto, Pelotas and São Luís). Ribeirão Preto 2016/2017, Pelotas 2015/2016, and São Luís 2016.

Variable	*n*	%
Age in years		
18	1640	24.8
19	753	11.4
21	134	2.0
22	3241	48.9
23	852	12.9
Sex		
Male	3060	46.2
Female	3560	53.8
Socioeconomic class *		
A	507	7.7
B	2521	38.1
C	2737	41.3
D/E	384	5.8
Unknown	471	7.1
Current smoking		
No	5901	89.1
Yes	719	10.9
Current alcohol consumption		
No	2356	35.6
Yes	4060	61.3
Unknown	204	3.1
Physical activity		
Insufficiently active	2424	36.6
Active	4196	63.4
Total	6620	100.0

Legend: * Mean monthly household income of each socioeconomic class in Brazil in 2016: A = USD 5989.00, B = USD 2021.00, C = USD 1241.00, D/E = US$ 220.00.

**Table 2 nutrients-15-00324-t002:** Mean and standard deviation of bone mineral density and consumption of sugar-sweetened beverages by young people of the RPS Brazilian Birth Cohorts Consortium (Ribeirão Preto, Pelotas, and São Luís). Ribeirão Preto 2016/2017, Pelotas 2015/2016, and São Luís 2016.

Variable	*n*	Minimum	Maximum	Mean	SD
Total body BMD (g/cm^2^)	6620	0.7	1.7	1.2	0.1
Lumbar spine BMD (g/cm^2^)	6620	0.6	1.8	1.2	0.1
Daily SSB consumption frequency	6620	0	16.7	1.6	1.7
1st tertile	2253	0	0.6	0.3	0.3
2nd tertile	2179	0.7	2.0	1.2	0.6
3rd tertile	2188	2.1	16.7	3.4	2.0
Daily SSB consumption (mL/day)	6620	0	3862.5	281.5	374.1
1st tertile	2212	0	83.2	39.9	34.7
2nd tertile	2205	83.4	275.5	176.6	82.1
3rd tertile	2203	275.6	3862.5	629.0	471.5
Daily energy contribution of SSB (%)	6620	0	25.7	5.9	5.7
1st tertile	2207	0	2.4	1.5	1.5
2nd tertile	2207	2.5	6.8	4.8	2.6
3rd tertile	2206	6.9	25.7	11.4	6.2

Legend: BMD: bone mineral density; SSB: sugar-sweetened beverages; SD: standard deviation.

**Table 3 nutrients-15-00324-t003:** Association between the daily consumption frequency of sugar-sweetened beverages and bone mineral density in young people of the RPS Brazilian Birth Cohorts Consortium (Ribeirão Preto, Pelotas, and São Luís). Ribeirão Preto 2016/2017, Pelotas 2015/2016, and São Luís 2016.

Daily Consumption Frequency			Total Body Bone Mineral Density				Lumbar Spine Bone Mineral Density			
Tertile	*n*	Number of	*β* ^a^	95% CI	*p*	Standardized	*β* ^a^	95% CI	*p*	Standardized
		Times				*β* ^b^				*β* ^b^
Unadjusted analysis										
1st	2253	0 to 0.6	Ref.	-	-	-	Ref.	-	-	-
2nd	2179	0.7 to 2.0	0.005	−0.002; 0.011	0.159	0.02	0	−0.007; 0.008	0.913	0.001
3rd	2188	2.1 to 16.7	−0.003	−0.009; 0.004	0.406	−0.120	−0.009	−0.017; −0.001	0.032	−0.030
Adjusted analysis ^c^										
1st	2253	0 to 0.6	Ref.	-	-	-	Ref.	-	-	-
2nd	2179	0.7 to 2.0	−0.001	−0.007; 0.005	0.689	−0.005	−0.001	−0.009; 0.007	0.781	0.004
3rd	2188	2.1 to 16.7	−0.009	−0.014; −0.003	0.005	−0.036	−0.012	−0.021; −0.004	0.003	−0.043
Adjusted analysis ^d^										
1st	2253	0 to 0.6	Ref.	-	-	-	Ref.	-	-	-
2nd	2179	0.7 to 2.0	−0.001	−0.005; 0.005	0.971	−0.001	−0.000	−0.008; 0.008	0.975	−0.001
3rd	2188	2.1 to 16.7	−0.004	−0.009; 0.001	0.131	−0.017	−0.008	−0.016; −0.001	0.036	−0.030

Legend: CI: confidence interval. ^a^ β: linear regression coefficient. ^b^ Standardized β: effect size. ^c^ Analysis adjusted for sex, socioeconomic class, alcohol consumption, smoking, and physical activity. ^d^ Analysis adjusted for sex, socioeconomic class, alcohol consumption, smoking, physical activity, and body mass index.

**Table 4 nutrients-15-00324-t004:** Association between the daily consumption of sugar-sweetened beverages and bone mineral density in young people of the RPS Brazilian Birth Cohorts Consortium (Ribeirão Preto, Pelotas, and São Luís). Ribeirão Preto 2016/2017, Pelotas 2015/2016, and São Luís 2016.

Daily Consumption (mL/day)			Total Body Bone Mineral Density				Lumbar Spine Bone Mineral Density			
Tertile	*n*	mL/day	*β* ^a^	95% CI	*p*	Standardized *β* ^b^	*β* ^a^	95% CI	*p*	Standardized *β* ^b^
Unadjusted analysis										
1st	2212	0 to 83.2	Ref.	-	-	-	Ref.	-	-	-
2nd	2205	83.4 to 275.5	0.006	−0.001; 0.013	0.073	0.025	−0.002	−0.010; 0.006	0.647	−0.006
3rd	2203	275.6 to 3862.5	0.008	−0.002; 0.015	0.072	0.035	0.001	−0.007; 0.009	0.726	0.005
Adjusted analysis ^c^										
1st	2212	0 to 83.2	Ref.	-	-	-	Ref.	-	-	-
2nd	2205	83.4 to 275.5	−0.001	−0.007; 0.005	0.793	−0.003	−0.003	−0.011; 0.005	0.477	−0.010
3rd	2203	275.6 to 3862.5	−0.004	−0.011; 0.002	0.159	−0.018	−0.004	−0.012; 0.004	0.362	−0.014
Adjusted analysis ^d^										
1st	2212	0 to 83.2	Ref.	-	-	-	Ref.	-	-	-
2nd	2205	83.4 to 275.5	−0.001	−0.006; 0.004	0.778	−0.003	−0.003	−0.011; 0.005	0.474	−0.010
3rd	2203	275.6 to 3862.5	−0.003	−0.008; 0.002	0.286	−0.012	−0.003	−0.011; 0.005	0.488	−0.010

Legend: CI: confidence interval. ^a^ β: linear regression coefficient. ^b^ Standardized β: effect size. ^c^ Analysis adjusted for sex, socioeconomic class, alcohol consumption, smoking, and physical activity. ^d^ Analysis adjusted for sex, socioeconomic class, alcohol consumption, smoking, physical activity, and body mass index.

**Table 5 nutrients-15-00324-t005:** Association between the daily energy contribution of sugar-sweetened beverages and bone mineral density in young people of the RPS Brazilian Birth Cohorts Consortium (Ribeirão Preto, Pelotas, and São Luís). Ribeirão Preto 2016/2017, Pelotas 2015/2016, and São Luís 2016.

Daily Energy Contribution (%)			Total Body Bone Mineral Density				Lumbar Spine Bone Mineral Density			
Tertile	*n*	%	*β* ^a^	95% CI	*p* Value	Standardized	*β* ^a^	95% CI	*p*	Standardized
						*β* ^b^				*β* ^b^
Unadjusted analysis										
1st	2207	0 to 2.4	Ref.	-	-	-	Ref.	-	-	-
2nd	2207	2.5 to 6.8	0.006	−0.000; 0.013	0.052	0.028	0.001	−0.007; 0.008	0.877	0.002
3rd	2206	6.9 to 25.7	0.004	−0.002; 0.011	0.202	0.018	−0.001	−0.009; 0.007	0.84	−0.003
Adjusted analysis ^c^										
1st	2207	0 to 2.4	Ref.	-	-	-	Ref.	-	-	-
2nd	2207	2.5 to 6.8	0	−0.006; 0.006	0.929	−0.001	0	−0.008; 0.008	0.937	0.001
3rd	2206	6.9 to 25.7	0.002	−0.018; 0.004	0.46	−0.010	−0.002	−0.010; 0.006	0.66	−0.006
					Adjusted analysis ^d^					
1st	2207	0 to 2.4	Ref.	-	-	-	Ref.	-	-	-
2nd	2207	2.5 to 6.8	0	−0.005; 0.006	0.835	0.002	0.001	−0.007; 0.009	0.804	0.003
3rd	2206	6.9 to 25.7	−0.002	−0.007; 0.003	0.415	−0.009	−0.002	−0.010; 0.006	0.645	−0.006

Legend: CI: confidence interval. ^a^ β: linear regression coefficient. ^b^ Standardized β: effect size. ^c^ Analysis adjusted for sex, socioeconomic class, alcohol consumption, smoking, and physical activity. ^d^ Analysis adjusted for sex, socioeconomic class, alcohol consumption, smoking, physical activity, and body mass index.

## Data Availability

The data that support the findings of this study are available from the authors by writing to rosangela.flb@ufma.br, hdgs.epi@gmail.com, or hbettiol@fmrp.usp.br, but restrictions apply to the availability of these data, which were used under license for the current study and so are not publicly available. Data are however available from the authors upon reasonable request and with permission of Rosangela Fernandes Lucena Batista or Helen Gonçalves or Heloisa Bettiol.

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
