# Peer review of "High Consumption of Sugar-Sweetened Beverages Is Associated with Low Bone Mineral Density in Young People: The Brazilian Birth Cohort Consortium"

_nutrients, 2023, doi:10.3390/nu15020324_

Round 1
Reviewer 1 Report
Dear Authors,
First of all, I would like to thank you for this study
In the abstract please add the among Brazilian young people.
The introduction is short please add the justification. Furthermore, I would like to know if there were previous studies on the prevalence.
Is there a difference in the region's lifestyles?
Methods. are clear.
Results presented in sex-specific due to differences in gender might be useful.
Discussions are clear.
In the limitation, I would suggest adding the geographic region and family history.
References also include recent articles in the field.
Author Response
Dear Reviewer,
We would like to thank you for reviewing and commenting on our article. Suggested changes are answered separately.
The following is a detailed description of how we addressed reviewers’ comments and suggestions.
- The introduction is short please add the justification. Furthermore, I would like to know if there were previous studies on the prevalence.
We appreciate the reviewer’s comment. We changed it as requested (lines 75-78).
- Is there a difference in the region's lifestyles?
Yes, there are some differences in terms of lifestyles among cities, especially based on geographic location and climate. São Luis is a metropolitan city on the cost (Atlantic Oceano) in the northeast of Brazil and has a tropical climate. Ribeirao Preto is in the southwest of Brazil, and the average temperature of the city throughout the year is 23 oC (73 oF). Pelotas is a city in the extreme south of Brazil with a winter season with a temperature ranging from -5 to 15 oC. However, as we described in the statistical section, the BMD tests of interactions between city and SSB and between sex and SSB on total body and lumbar spine were tested. These interactions were not significant and therefore included in the same model (city and sex). In addition, the level of physical activity was also included in the adjustment model.
- are clear.
We appreciate the reviewer’s comment.
- Results presented in sex-specific due to differences in gender might be useful.
We appreciate the reviewer’s comment.
- Discussions are clear.
We appreciate the reviewer’s comment.
- In the limitation, I would suggest adding the geographic region and family history.
Thank you for your critical point. The paragraph below was added in the Limitation paragraph:
Other limitation is the geographical difference of the study pospulation. São Luis is a metropolitan city in the cost in the northeast of Brazil and has a tropical climate, Ribeirao Preto is in the southwest with an average temperature of 23 ° C throughout the year, and Pelotas is in the extreme south of the country with a winter season with temperature ranging from -5 to 15 oC. Considering that, the city was considered on adjustments to control for geographical differences. Another limitation refers to a family history of bone health, such as osteoporosis.
- References also include recent articles in the field.
We appreciate the reviewer’s comment.

Reviewer 2 Report
The manuscript evaluates the associations between consumption of sugar-sweetened beverages and bone mineral density in young people. Overall, the data is interesting.
1. Methods: the author stated that “Femoral measurements were not used since this
bone continues to grow until 20 years of age”. For whole body and lumbar spine, have the bone minerals reached their peak values in the targeted population? This statement maybe not correct.
2. Methods: why did the author use tertiles but not continuous variables in the linear regression model? By using continuous variables, the decreasement of BMD for per unit or per SD of exposures could be evaluated. Ancova analysis can be applied to evaluate the group differences among different tertiles.
3. Results: three significant digits are recommended for BMD data.
4. Results: for beta coefficient and 95%CI, only 1 significant digit were listed, which might not present the effect size accurately.
5. Discussion: the authors applied three assessments of SSB intakes to evaluate their associations with bone health. The results showed that the frequency but not the total amount of SSB correlated with BMD, why? This is suggested to be discussed in more detail.
6. minor concerns:
1) Introduction: references should be cited in line 42-46 and 63-67.
2) line 164:”until 20 years of age 2”. 2 should be deleted
3)Table 4, the ranges for each tertile are similar.
4) other errors should be corrected: e,g “,3560” in table 1.
Author Response
Dear Reviewer,
We would like to thank you for reviewing and commenting on our article. Suggested changes are answered separately.
The following is a detailed description of how we addressed reviewers’ comments and suggestions.
Reviewer 2
- Methods: the author stated that “Femoral measurements were not used since this bone continues to grow until 20 years of age”. For whole body and lumbar spine, have the bone minerals reached their peak values in the targeted population? This statement maybe not correct.
We appreciate the reviewer’s comment. We changed it as requested. The paragraph below was added in the methods.
Femoral measurements were not used since this bone continues to grow until 20 years of age and there was age difference among cohorts.
- Methods: why did the author use tertiles but not continuous variables in the linear regression model? By using continuous variables, the decreasement of BMD for per unit or per SD of exposures could be evaluated. Ancova analysis can be applied to evaluate the group differences among different tertiles.
We appreciate the reviewer’s comment. The analysis was carried out considering the BBA tertiles in order to assess the consumption of these drinks by young people in smaller amounts, moderately and in larger amounts. We don’t use continuous variables for these reasons. However, changes or decreasement in BMD standard deviation were checked using effect size analysis (please check explanations in the lines 231-235).
- Results: three significant digits are recommended for BMD data.
We appreciate the reviewer’s comment. We changed it as requested (table 3).
- Results: for beta coefficient and 95%CI, only 1 significant digit were listed, which might not present the effect size accurately.
We appreciate the reviewer’s comment. We changed it as requested (table 4).
- Discussion: the authors applied three assessments of SSB intakes to evaluate their associations with bone health. The results showed that the frequency but not the total amount of SSB correlated with BMD, why? This is suggested to be discussed in more detail.
We discussed these aspects in the second paragraph of the discussion section:
“We found no associations between the frequency of daily SSB consumption and total body BMD or between the amount consumed and energy contribution of SSB and total body or lumbar spine BMD. However, despite the small magnitude of the effect, a higher frequency of SSB consumption was associated with lower lumbar spine BMD. A change of one standard deviation in the SSB consumption frequency reduced the lumbar spine BMD by 0.03 standard deviations. The high consumption of SSB by these adolescents probably did not result in marked bone demineralization because this is a young population in which bone mineralization is still expected to occur 35. This fact may also explain the lack of associations between the frequency of daily SSB consumption and total body BMD and between the amount consumed and energy contribution of SSB and total body and lumbar spine BMD. Furthermore, these results are important because the observation of an association only with lumbar spine BMD may indicate the onset of bone demineralization that could spread throughout the body if the excessive consumption of SSB continues in the sample studied over the years.”
- minor concerns:
1) Introduction: references should be cited in line 42-46 and 63-67.
We appreciate the reviewer’s comment. We changed it as requested.
2) line 164:”until 20 years of age 2”. 2 should be deleted
We appreciate the reviewer’s comment. We changed it as requested.
3)Table 4, the ranges for each tertile are similar.
We appreciate the reviewer’s comment. We changed it as requested.
4) other errors should be corrected: e,g “,3560” in table 1.
We appreciate the reviewer’s comment. We changed it as requested.